# Prevalence of Ocular Demodicosis and Ocular Surface Conditions in Patients Selected for Cataract Surgery

**DOI:** 10.3390/jcm9103069

**Published:** 2020-09-23

**Authors:** Katarzyna Nowomiejska, Piotr Lukasik, Agnieszka Brzozowska, Mario Damiano Toro, Aleksandra Sedzikowska, Katarzyna Bartosik, Robert Rejdak

**Affiliations:** 1Department of General Ophthalmology, Medical University of Lublin, 20-079 Lublin, Poland; piotrlukasik.med@gmail.com (P.L.); toro.mario@email.it (M.D.T.); robert.rejdak@umlub.pl (R.R.); 2Department of Ophthalmology, John Paul II Public Hospital in Zamosc, 22-400 Zamosc, Poland; 3Department of Mathematics and Medical Biostatistics, Medical University of Lublin, 20-090 Lublin, Poland; agnieszka.brzozowska@umlub.pl; 4Faculty of Medicine, Collegium Medicum Cardinal Stefan Wyszynski University, 01-815 Warsaw, Poland; 5Department of General Biology and Parasitology, Medical University of Warsaw, 02-004 Warsaw, Poland; aleksandra.sedzikowska@wum.edu.pl; 6Chair and Department of Biology and Parasitology, Medical University of Lublin, 20-080 Lublin, Poland; katarzyna.bartosik@umlub.pl

**Keywords:** *Demodex folliculorum*, ocular demodicosis, cataract surgery, Schirmer I test, break up time test, blepharitis, tear film

## Abstract

The aim of the study was to analyze the prevalence of ocular demodicosis and ocular surface conditions in patients selected for cataract surgery. Eyelashes from 73 patients selected for cataract surgery were evaluated at ×40 and ×100 magnification using light microscopy. The anterior segment was assessed with the slit lamp. Additionally, Schirmer I and break up time (BUT) tests were carried out before surgery and 1 and 3 months postoperatively. A specially designed questionnaire containing e.g., information about chronic skin and eye diseases, previous ophthalmic surgeries, and patient’s hygiene habits was used to assess the demographic variables. A majority of patients were at the age of 70–79 years, and there were more females (83%) in the study group. *Demodex folliculorum* was found in 48% of the patients. There was a correlation between the number of parasites and the presence of blepharitis, discharge at eyelid margins, and conjunctival hyperemia. Schirmer I and BUT test results were lower in patients with *Demodex* infestation before and after cataract surgery. The higher number of mites was correlated with lower Schirmer I test results postoperatively. The presence of *Demodex* mites influences the conjunctiva and lid margins leading to inflammation. The higher number of *Demodex* mites disturbs the tear film over time after cataract surgery.

## 1. Introduction

*Demodex* spp. is an obligatory ectoparasite of hair follicles and sebaceous glands in humans of different ethnic groups and other mammals. The first description was given by Simon in 1843 [1]. Its lifespan is supposed to be up to 3 weeks from the egg stage to the adult stage [2]. Demodicosis is a condition caused by the presence of *Demodex* species. Currently, more than 100 species of *Demodex* have been described in literature but only two of them, *Demodex folliculorum* (Simon, 1842) and *Demodex brevis* (Akbulatova 1963), are human parasites living in Meibomian glands of the skin and Meibomian glands and the follicles of eyelashes [3,4]. The most prevalent is *D. folliculorum* (Figure 1), but *D. brevis* may be also found in the same host [5,6]. The adult *D. folliculorum* stages have a length of 279–294 μm and 104 μm × 41 μm arrowhead-shaped ova, whereas *D. brevis* is smaller (165–208 µm) [7]. *Demodex* sp. infestation is associated with acne vulgaris, rosacea, and seborrheic dermatitis [8]. Ophthalmic demodicosis manifests as persistent blepharitis [9], chalazion, and dry eye syndrome [10] or may be associated with eyelid basal cell carcinoma [11]. The ocular invasion of *Demodex* may be asymptomatic; however, when it results in blepharitis, the symptoms vary from being a chronic condition, dry eye, to a severe compromise of the ocular surface with a morbid impact on patients’ quality of life [12].

Cataract surgery is the most prevalent surgery performed in ophthalmology, but it is also considered as an ocular surface damaging event. It is also known that the incidence and severity of dry eye symptoms may increase after cataract surgery [13]. To achieve the best outcome in cataract surgery, a healthy ocular surface is crucial. Patients with more severe ocular surface disease are at higher risk of post-operative complications such as secondary infections.

The aim of the study was to determine the prevalence of *D. folliculorum* in eyelash follicles of patients selected for cataract surgery and its relationship with eye symptoms and related ocular surface condition.

## 2. Material and Methods

### 2.1. Experimental Procedures

The study included 73 consecutive patients that underwent routine cataract surgery in the period from July 2018 to December 2018. Each patient had one eye examined before the surgery and after one and three months postoperatively at the Department of Ophthalmology, John Paul II Public Hospital in Zamość, Poland.

The cataract surgeries were performed by the same experienced surgeon in a standard manner after topical anesthesia with proparacaine hydrochloride 0.5%. After making a 2.2 mm clear corneal incision, continuous capsulorrhexis, hydrodyssection, and phacoemulsification were performed (Infiniti Vision System Alcon, Fort Worth, Texas, USA) and the IOL was inserted into the capsular bag. All surgeries were performed without complications.

The postoperative standard care included application of topical antibiotic drops (Oftaquix, Santen Oy, Tampere, Finland) five times per day for two weeks and steroid drops (Dexafree, Santen Oy) five times per day for two weeks, then reduced to three times per day for another two weeks.

Ten eyelashes were epilated from one eye of each subject before cataract surgery with the use of sterile laboratory tweezers, placed on light microscope slides at ×40 and ×100 magnification (Delta Optical Genetic Pro Bino, Minsk Mazowiecki, Poland), and examined to determine the presence and quantity of mites. A sample was considered positive if at least one parasite was found [14].

A specially designed questionnaire containing demographic (age, gender, job or faculty, place of residence) and clinical data (history of chronic dermatological and ocular diseases and patient’s hygiene habits) was completed for each participant based on the anamnesis preoperatively. No special treatment regimen was applied in regard to the *Demodex* infestation.

The patients were examined with the use of the slit lamp before surgery and postoperatively after one and three months. The following parameters of the anterior segment were checked preoperatively: hyperemia of the conjunctiva, blepharitis (teleangiectasia of the lid margin), loss of lashes, discharge on the lid margins, and defects of epithelium of the cornea.

Additionally, the Schirmer I test (without anesthesia with the eyes closed for 2 or 5 min.) and the tear film break up time (BUT) test were assessed at each visit (preoperatively, after one and three months postoperatively) both in the group with *Demodex* infestation and without *Demodex* infestation (control group).

The study was performed in accordance with the Declaration of Helsinki. The study was approved by the Ethics Committee of the Medical University of Lublin (number of approval KE-0254/135/2018). All participants provided their written informed consent to the study.

### 2.2. Data Analysis

Statistical analysis was performed using STATISTICA 13.0 (StatSoft, Krakow, Poland) software. The Mann-Whitney U test and Kruskal-Wallis test were used to compare two independent groups. Spearman’s R correlation was used to assess the relationship between the variables. A *p* value less than 0.05 was considered to be of statistical significance.

## 3. Results

There were 61 (83%) females and 12 (17%) males in the study group; 52% (*n* = 38) lived in the city and 48% (*n* = 35) in the country. Most of the patients were at the age of 70–79 years (49.32%, *n* = 36), 15.07% (*n* = 11) were at the age of 80–89 years, 28.77% (*n* = 21) were at the age of 60–69 years, 2.74% (*n* = 2) were at the age of 40–49 years, 2.74% (*n* = 2) were at the age of 50–59 years, and 1.36% (*n* = 1) were older than 90 years.

The prevalence of *D. folliculorum* infestation was found to be 48.0% of all studied participants. The mean number of *Demodex* individuals found in one eye was 1.3 (range 0–11). In our study, there was no significant relationship between the *Demodex* infestation and the age (Chi^2^ = 6.45; *p* = 0.17), gender (Chi^2^ = 0.88; *p* = 0.64), education (Chi^2^ = 4.52; *p* = 0.34), and place of residence of the patients (Chi^2^ = 1.63; *p* = 0.44).

The most frequent chronic diseases in the subjects were: cardiovascular diseases (73.97%), eye diseases (43.84%), metabolic diseases (35.62%), and 58.9% of patients suffered from more than one chronic disease, however, no statistically significant correlation was found between demodicosis and the coexisting one or more chronic disease in the studied group (Chi^2^ = 0,40; *p* = 0,98).

In the group of patients with the confirmed presence of *Demodex*, patients who reported using artificial tears (Systane, Alcon, Fort Worth, TX, USA) and tissues (Blephaclean, Thea, Cedex Clermont-Ferrand, France) for cleaning the lid margins were found to have greater numbers of mites: more than two mites were found in 37% of these patients and only in 6% who were not using tears or tissues (*p* < 0.00001) (Table 1).

The most common finding in the slit lamp examination was the discharge on the lid margins (41.10%), lack of lashes (38.36%), hyperemia of the conjunctiva (36.99%), blepharitis (34.25%), and defects of the corneal epithelium (1.37%). There was a significant correlation between the presence of blepharitis and the number of *Demodex* mites (*p* = 0.0006). More than two mites were present in 40% of patients with blepharitis and in 6% of patients without blepharitis (Table 2).

There was a significant correlation between the presence of conjunctival hyperemia and the number of *Demodex* mites (*p* = 0.005): more than two mites were present in 33% of patients with conjunctival hyperemia and in 15% of patients without conjunctival hyperemia (Table 3).

A significant correlation was found between the presence of discharge and the number of *Demodex* mites (*p* = 0.002). More than two mites were present in 33% of patients with discharge and only in almost 7% of patients without discharge (Table 4). There was no significant correlation between the number of mites and the lack of lashes and corneal changes (Table 5 and Table 6).

There were significant differences both in Schirmer I (Table 7) and BUT test (Table 8) between patients with *Demodex* infestation and without. Schirmer test results were lower in patients with Demodex infestation. BUT test was significantly shorter in *Demodex* positive patients.

The statistical analysis did not reveal any significant differences in the Schirmer I and BUT test results between the examinations pre- and postoperatively both in the group with *Demodex* infestation and without (Table 7 and Table 8).

A correlation was found between the preoperative number of mites and the results of the Schirmer I test after one month (R = −0.24, *p* = 0.04) and after 3 months (R = −0.25, *p* = 0.03) (Table 9). The higher number of mites was correlated with lower Schirmer I test results.

There was also a significant relationship between the results of BUT and the number of mites after one month (R = −0.31, *p* = 0.01) and 3 months (R = −0.26, *p* = 0.02) postoperatively (Table 10).

## 4. Discussion

In recent years, the *Demodex* infestation has become an increasing public health concern. Mites can be found in all human races around the world [9,15,16,17,18]. *D. folliculorum* occurs more frequently than *D. brevis* and infestation by both species increases with age [19]. In our study, only *D. folliculorum* was found in all positive samples. The possible reason may be that *D. folliculorum* can be more easily isolated than *D. brevis* [20], as *D. folliculorum* exists in the lash follicle, whereas *D. brevis* penetrates deeper into the lash’s sebaceous gland and the meibomian gland [2]. Thus, *D. folliculorum* is more commonly seen in posterior blepharitis, or keratoconjunctivitis and *D. brevis* is more common in the sebaceous gland- or meibomian gland-related diseases, such as chalazion [15].

*Demodex* can induce inflammation of the skin and lid margin, Meibomian gland dysfunction, blepharoconjunctivitis, and blepharokeratitis. [2]. Intensive *D. folliculorum* invasions cause keratinization, hyperplasia, distension, and melanocyte aggregation. Large populations of *D. brevis* may destroy glandular cells, produce granuloma, and plug the ducts of the Meibomian or sebaceous glands [21]. However, the relevance of *Demodex* spp. in blepharitis is still controversial. Most authors demonstrate a higher prevalence of *Demodex* mites in patients with blepharitis compared to healthy controls [9,22,23], which in accordance with our study, whereas some authors show a similar prevalence of *Demodex* mites in blepharitis and control groups. Kemal found *Demodex* in 28.8% (49/170) of patients with blepharitis and in 26.7% (88/330) of controls [24]. The difference between the two groups was not statistically significant and there was no relationship between the presence of *D. folliculorum* and host factors (age, sex).

In turn, Sedzikowska et al. examined 134 patients with blepharitis and 76% had positive result for *Demodex*. The authors also found that the sex of the subjects was not a factor conducive to infection, but their age was positively correlated with the risk of infection [25]. In our study, the result was very similar, as 72% of the patients with blepharitis had positive result for the presence of *Demodex* but there was no significant relationship of the *Demodex* infestation with age and gender. It may be due to the fact that a majority of our patients were at the age of 70–79 years, and there were more females than males in the study group, this profile is typical for patients who undergone cataract surgery.

The literature suggests a correlation between *Demodex* mites and cylindrical dandruff [26] or loss of eyelashes and trichiasis [27,28]. In our study, there was no significant correlation between the number of mites and lack of lashes.

Severe lid margin inflammation can be a result of mechanical blockage and delayed host immune hypersensitive reaction [2]. Inflammation of the lid margin can lead to inflammation of the conjunctiva [29]. Moreover, mites may be a vector for bacteria in the eye causing conjunctivitis [30].

According to the literature data, immunosuppression is an important predisposing factor for development of symptomatic *Demodex* spp. invasion [31,32,33,34]. On the other hand, the research conducted by Kosik-Bogacka in the group of patients with haematologic diseases did not show any significant differences between the prevalence of *D. folliculorum* in the study group and in the control group [35].

The risk of the occurrence of ocular symptoms in patients increases with the rise in the density of *Demodex* mites in one sample [36,37,38,39], but a majority of infestation cases seem to be asymptomatic [7]. As humans are the only host of *D. folliculorum and D. brevis* mites, no animal models of ocular demodicosis have been successfully established [37]. No previous research has demonstrated whether a minimal number of mites must be present to cause symptoms. As demonstrated by our results, in the case of blepharitis, hyperemia, and discharge, more than two parasites were found in one sample.

It is known that Meibomian gland dysfunction and deterioration of the tear film may increase after cataract surgery [40,41,42]. However, the exact mechanism by which cataract surgery impairs the Meibomian gland function remains unknown. Lee et al. found that an increasing number of *Demodex* reduced the BUT but did not affect the results of the Schirmer test [43].

In our study there were significant differences in the Schirmer I and BUT tests between patients with *Demodex* infestation and without in the 3-month postoperative period. Thus, cataract surgery impaired significantly the tear film and homeostasis of ocular surface in patients with *Demodex* infestation. It is already known that there is a substantial decrease in the BUT test approximately 3 months after cataract surgery [44], thus Demodex infestation can even exaggerate the symptoms of dry eye syndrome after cataract surgery. Patients should be examined before cataract surgery in regard to *Demodex* infestation and informed about dry symptoms that can evolve postoperatively.

The presence of *Demodex* mites in hair follicles may cause dislocation of the base of the hair and excessive loss of eyelashes and eyebrows [45]. Mite eggs laid at the base of the lashes contribute to follicular distention and misdirected lashes [30]. In turn, epithelial hyperplasia and reactive hyperkeratinization are induced by microabrasions caused by the mite claws [46]. In our research, the number of *Demodex* mites was similar in the group of patients with and without the lack of lashes.

The reason may be that the number of lashes was not really counted, assessment of lack of lashes was done only subjectively by the examiner.

In our study, the prevalence of *Demodex* infestation in all patients (with and without blepharitis) was 48%, and most of the patients were at the age 70–79 years. It is relatively low, comparing to the results reported by other authors, e.g., Sedzikowska: 77% of infected patients in a group over 70 years [24], Czepita: 95% of patients aged 71–96 years [47], and Vargas-Arzola: 64% of patients aged 76–85 years [48]. In a study conducted by Post and coworkers, *Demodex* was observed in 84% of the general population aged 60 years and 100% of the general population aged above 70 years [38]. However, there are some studies with similar prevalence, for example 40.2% of patients suffering from ocular discomfort [49]. Lower prevalence of *Demodex* infestation in our study may be explained by the fact that patients selected for the cataract surgery take more attention to the lid hygiene, than normal population of patients at this age.

Demodicosis can be diagnosed by sampling eyelashes, which are then placed on a slide and observed under the light microscope. This method was used in our study to confirm the presence of *Demodex* mites. In vivo confocal laser scanning microscopy (CLSM) is an alternate method to confirm diagnosis [39].

The treatment of demodicosis is challenging, as demodicosis is a chronic condition requiring long-term therapy. The patients in our study who declared using tissues and lubricants for eye hygiene had a higher number of mites; however, it is hard to assess the real influence of this practice on the mite population, as the ingredients of these products and the frequency of their use were not validated. Possibly, the use of tissues and lubricants was prompted by the reaction to the symptoms caused by the presence of the higher number of the mites in the patients. The question remains what number of mites represents normal infestation versus pathognomonic overgrowth or what number of mites is required to elicit symptoms [50].

## 5. Conclusions

*Demodex folliculorum* infestation is a common condition in patients selected for cataract surgery. The higher number of *Demodex* mites influences the conjunctiva and lid margin and leads to inflammation of ocular surface and disturbance of the tear film.

## Figures and Tables

**Figure 1 jcm-09-03069-f001:**
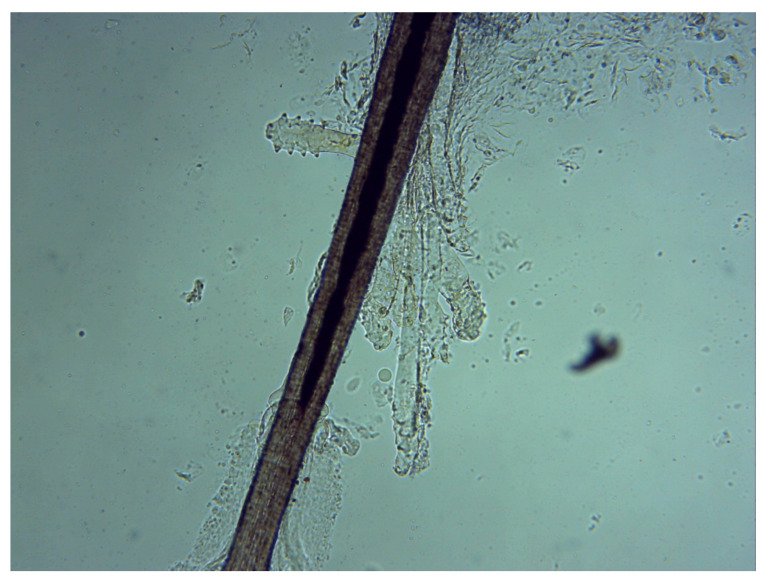
*Demodex folliculorum* on an eyelash follicle—5 adults and 2 larval forms of (original magnification 100×) (photograph by Renata Przydatek-Tyrajska).

**Table 1 jcm-09-03069-t001:** Relationship between the number of parasites found in one sample and the presence of blepharitis.

Artificial Tears or Tissues	Number of *Demodex* Mites			Total *n* (%)
	0 *n* (%)	1–2 *n* (%)	>2 *n* (%)	
Yes	0	17	10	27
	0.00%	62.96%	37.04%	100.00%
No	38	5	3	46
	82.61%	10.87%	6.52%	100.00%
Total	38	22	13	73
	52.05%	30.14%	17.81%	100.00%
*Chi^2^ = 46.52; p < 0.00001 **				

* The results of statistical analysis.

**Table 2 jcm-09-03069-t002:** Relationship between the number of parasites found in one sample and the presence of blepharitis.

Blepharitis	Number of *Demodex* mites			Total *n* (%)
	0 *n* (%)	1–2 *n* (%)	>2 *n* (%)	
Absent	31	14	3	48
	(64.58)	(29.17)	(6.25)	(100.00)
Present	7	8	10	25
	(28.00)	(32.00)	(40.00)	(100.00)
Total	38	22	13	73
	(52.05)	(30.14)	(17.81)	(100.00)
*Chi^2^ = 14.78; p = 0.0006 **				

* The results of statistical analysis.

**Table 3 jcm-09-03069-t003:** Relationship between the number of parasites found in one sample and the presence of conjunctival hyperemia.

Conjunctival Hyperemia	Number of *Demodex* Mites			Total *n* (%)
	0 *n* (%)	1–2 *n* (%)	>2 *n* (%)	
Absent	30	12	4	46
	(65.22)	(26.08)	(8.70)	(100.00)
Present	8	10	9	27
	(29.63)	(37.04)	(33.33)	(100.00)
Total	38	22	13	73
	(52.05)	(30.14)	(17.81)	(100.00)
*Chi^2^ = 10.62; p = 0.005 **				

* The results of statistical analysis.

**Table 4 jcm-09-03069-t004:** Relationship between the number of parasites found in one sample and the presence of discharge.

Discharge at Eyelid Margins	Number of *Demodex* Mites			Total *n* (%)
	0 *n* (%)	1–2 *n* (%)	>2 *n* (%)	
Absent	29	11	3	43
	(67.44)	(25.58)	(6.98)	(100.00)
Present	9	11	10	30
	(30.00)	(36.67)	(33.33)	(100.00)
Total	38	22	13	73
	(52.05)	(30.14)	(17.81)	(100.00)
*Chi^2^ = 12.37; p = 0.002 **				

* The results of statistical analysis.

**Table 5 jcm-09-03069-t005:** Relationship between the number of parasites found in one sample and the lack of lashes.

Lack of Lashes	Number of *Demodex* Mites			Total *n* (%)
	0 *n* (%)	1–2 *n* (%)	>2 *n* (%)	
Absent	27	13	5	45
	(60.00)	(28.89)	(11.11)	(100.00)
Present	11	9	8	28
	(39.29)	(32.14)	(28.57)	(100.00)
Total	38	22	13	73
	(52.05)	(30.14)	(17.81)	(100.00)
*Chi^2^ = 4.44; p = 0.11 **				

* The results of statistical analysis.

**Table 6 jcm-09-03069-t006:** Relationship between the number of parasites found in one sample and corneal changes.

Corneal Changes	Number of *Demodex* Mites			Total *n* (%)
	0 *n* (%)	1–2 *n* (%)	>2 *n* (%)	
Absent	38	22	12	72
	(52.78)	(30.55)	(16.67)	(100.00)
Present	0	0	1	1
	(0.00)	(0.00)	(100.00)	(100.00)
Total	38	22	13	73
	52.05%	30.14%	17.81%	100.00%
*Chi^2^ = 4.68; p = 0.10 **				

* The results of statistical analysis

**Table 7 jcm-09-03069-t007:** Values of Schirmer test I results (mm) during the follow-up in a group of patients with *Demodex* infestation and without.

Visit	Demodex			No Demodex			Statistical Analysis	
	Mean	Median	SD	Mean	Median	SD	Z	*p*
Preoperatively	12.10	10.50	8.43	16.76	16.25	9.37	2.07	0.04 *
After one month	11.59	9.50	9.40	16.09	13.50	9.43	2.25	0.02 *
After 3 months	12.03	10.50	7.98	16.12	14.25	7.90	2.32	0.02 *

SD; standard deviation; * means statistical significance.

**Table 8 jcm-09-03069-t008:** Values of the BUT test results (sec) during the follow-up in a group of patients with *Demodex* infestation and without.

Visit	Demodex			No Demodex			Statistical Analysis	
	Mean	Median	SD	Mean	Median	SD	Mean	Median
Preoperatively	7.46	5.50	4.29	9.96	9.50	5.56	2.06	0.04 *
After one month	6.70	5.50	4.17	8.32	8.00	3.62	2.37	0.02 *
After 3 months	6.21	5.00	4.46	7.28	7.00	3.24	1.92	0.05

SD; standard deviation; * means statistical significance.

**Table 9 jcm-09-03069-t009:** Correlation between the number of mites and the results of the Schirmer I test at each visit (preoperatively, postoperatively after one month and after 3 months).

Visit	R *	*p*
Preoperatively	−0.22	0.06
After one month	−0.24	0.04
After 3 months	−0.25	0.03

* Spearman’s R coefficient.

**Table 10 jcm-09-03069-t010:** Correlation between the number of mites and the results of the BUT test at each visit (preoperatively, postoperatively after one month and after 3 months).

Visit	R *	*p*
Preoperatively	−0.30	0.01
After one month	−0.31	0.01
After 3 months	−0.26	0.02

* Spearman’s R coefficient.

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
