# Peer review of "Prevalence of Ocular Demodicosis and Ocular Surface Conditions in Patients Selected for Cataract Surgery"

_jcm, 2020, doi:10.3390/jcm9103069_

Round 1

Reviewer 1 Report

The study doesn't add relevant new information to the field of eyelid demodex infestation. Some of the results are controversial and not appropriately discussed (for example unaltered tear film status after cataract surgery). Moreover, standard care of patients after cataract surgery is not provided by the authors (antibiotic; steroid/non-steroid eyedrops - period of usage). These agents may alter the postoperative state of the ocular surface. The applied treatment in case of demodex infestation is also missing. What was the treatment regimen applied (if they were treated...)?

Author Response

The study doesn't add relevant new information to the field of eyelid Demodex infestation.

In the abstract (line 29-31), it is written:

There was a correlation between the number of parasites and the presence of blepharitis, discharge at eyelid margins, and conjunctival hyperemia.

And line 33-34: The presence of Demodex mites influences the conjunctiva and lid margins leading to inflammation.

Blepharitis is mentioned in the results section (line 153-155) as follows: The most common finding in the slit lamp examination was the discharge on the lid margins (41.10%), lack of lashes (38.36%), hyperemia of the conjunctiva (36.99%), blepharitis (34.25%), and defects of the corneal epithelium (1.37%).

Line 156-158: There was a significant correlation between the presence of blepharitis and the number of Demodex mites (p = 0.0006). More than two mites were present in 40% of patients with blepharitis and in 6% of patients without blepharitis (Table 2).

In the discussion chapter (line 305-334) it is written:

Demodex can induce inflammation of the skin and lid margin, Meibomian gland dysfunction, blepharoconjunctivitis, and blepharokeratitis. [2]. Intensive D. folliculorum invasions cause keratinization, hyperplasia, distension, and melanocyte aggregation. Large populations of D. brevis may destroy glandular cells, produce granuloma, and plug the ducts of the Meibomian or sebaceous glands [21]. However, the relevance of Demodex spp. in blepharitis is still controversial. Most authors demonstrate a higher prevalence of Demodex mites in patients with blepharitis compared to healthy controls [9,22,23], which in accordance with our study, whereas some authors show a similar prevalence of Demodex mites in blepharitis and control groups. Kemal found Demodex in 28.8% (49/170) of patients with blepharitis and in 26.7% (88/330) of controls [24]. The difference between the two groups was not statistically significant and there was no relationship between the presence of D. folliculorum and host factors (age, sex).

Some of the results are controversial and not appropriately discussed (for example unaltered tear film status after cataract surgery).

In the results section (line 209-293) it is now written:

There were significant differences both in Schirmer I (Table 7) and BUT test (Table 8) between patients with Demodex infestation and without. Schirmer test results were lower in patients with Demodex infestation. BUT test was significantly shorter in Demodex positive patients.

The statistical analysis did not reveal any significant differences in the Schirmer I and BUT test results between the examinations pre- and postoperatively both in the group with Demodex infestation and without (Table 7 and 8).

Table 7.

Values of Schirmer test I results (mm) during the follow-up in a group of patients with Demodex infestation and without (SD - standard deviation).

Visit

Demodex

No Demodex

Statistical analysis

Mean

Median

SD

Mean

Median

SD

Z

p

Preoperatively

12.10

10.50

8.43

16.76

16.25

9.37

2.07

0.04*

After one month

11.59

9.50

9.40

16.09

13.50

9.43

2.25

0.02*

After 3 months

12.03

10.50

7.98

16.12

14.25

7.90

2.32

0.02*

Table 8.

Values of the BUT test results (sec) during the follow-up in a group of patients with Demodex infestation and without (SD - standard deviation).

Visit

Demodex

No Demodex

Statistical analysis

Mean

Median

SD

Mean

Median

SD

Mean

Median

Preoperatively

7.46

5.50

4.29

9.96

9.50

5.56

2.06

0.04*

After one month

6.70

5.50

4.17

8.32

8.00

3.62

2.37

0.02*

After 3 months

6.21

5.00

4.46

7.28

7.00

3.24

1.92

0.05

A correlation was found between the number of mites and the results of the Schirmer I test after one month (R = -0.24, p = 0.04) and after 3 months (R = -0.25, p = 0.03) (table 9). The higher number of mites was correlated with lower Schirmer I test results.

Table 9. Correlation between the number of mites and the results of the Schirmer I test at each visit (preoperatively, postoperatively after one month and after 3 months)

Visit

R*

p

Preoperatively

-0.22

0.06

After one month

-0.24

0.04

After 3 months

-0.25

0.03

* Spearman's R coefficient

There was also a significant relationship between the results of BUT and the number of mites after one month (R = -0.31, p = 0.01) and 3 months (R = -0.26, p = 0.02) postoperatively (table 10).

Table 10. Correlation between the number of mites and the results of the BUT test at each visit (preoperatively, postoperatively after one month and after 3 months)

Visit

R

p

Preoperatively

-0.30

0.01

After one month

-0.31

0.01

After 3 months

-0.26

0.02

* Spearman's R coefficient

In the discussion chapter (line 353-359) it has been added:

In our study there were significant differences in the Schirmer I and BUT tests between patients with Demodex infestation and without in the 3-month postoperative period. Thus, cataract surgery impaired significantly the tear film and homeostasis of ocular surface in patients with Demodex infestation. It is already known that there is a substantial decrease in the BUT test approximately 3 months after cataract surgery [44], thus Demodex infestation can even exaggerate the symptoms of dry eye syndrome after cataract surgery. Patients should be examined before cataract surgery in regard to Demodex infestation and informed about dry symptoms that can evolve postoperatively.

Moreover, standard care of patients after cataract surgery is not provided by the authors (antibiotic; steroid/non-steroid eyedrops - period of usage). These agents may alter the postoperative state of the ocular surface.

In the Material and Methods section (line 90-92), the following sentence has been added:

The postoperative standard care included application of topical antibiotic drops (Oftaquix, Santen Oy) 5 times per  day for two weeks and steroid drops (Dexafree, Santen Oy) 5 times per day for two weeks, then reduced to 3 times per  day for another two weeks.

The applied treatment in case of demodex infestation is also missing. What was the treatment regimen applied (if they were treated...)?

In the Material and Methods section (line 99-100), the following sentence has been added: No special treatment regimen was applied in regard to the Demodex infestation.

Reviewer 2 Report

Title: Prevalence of ocular demodicosis and ocular surface conditions in patients selected for cataract surgery

In this study, the authors analyzed the prevalence of ocular demodicosis and ocular surface conditions in patients selected for cataract surgery. The results showed the presence of Demodex mites influences the conjunctiva and lid margins leading to inflammation, but does not disturb the tear film. Because a healthy ocular surface is crucial for the best outcome in cataract surgery, the influence of infection of Demodex mites on the ocular surface condition after cataract surgery is quite important. 

General comments:

  1. It is unclear what the novelty of this study is. The relationship between demodicosis and eye symptoms including ocular surface condition in normal (pre-operative) eyes has been previously reported. It may be important whether the demodicosis affects the eye condition after cataract surgery or not.
  2. It is unclear when ophthalmic examinations were performed. Although the authors described that there were no significant differences in the BUT and Schirmer I tests between examinations pre- and postoperatively in Abstract, no comparison between pre-operation and post-operation was described in Material and Methods and Results. To examine the influence of ocular demodicosis on the eye condition after cataract surgery, all results of ophthalmic examinations must be compared between pre-operation and post-operation. Additionally, comparison between patients with demodicosis and ones without demodicosis should be done before and after surgery, respectively.
  3. As the authors indicated, many studies reported deterioration of the tear film after cataract surgery. It is unclear why cataract surgery did not impair tear film and homeostasis of ocular surface in this study. If cataract surgery does not impair ocular surface regardless of the infection of Demodex mites, it is impossible to access the influence of demodicosis on the ocular surface condition after cataract surgery.
  4. Although the authors argued that the presence of Demodex mites does not disturb the tear film, the results in this study demonstrated the significant relationship between the number of mites and results of tear film test (Schirmer I test and BUT test). It is difficult to understand what the authors mean.
  5. The use of artificial tears and tissues should have a large impact on the results of this study. The patients in this study should be divided by the use of artificial tears and tissues.
  6. Descriptions in Discussions tend to repetition of the results. The authors should discuss the meaning of the results.

Specific comments:

  1. In lines 31-32, the comparison between pre-operation and post-operation is not described in Material and Methods and Results.
  2. In lines 32-33, the results (lines 138-142) suggested that infection of Demodex mites impaired the tear film.
  3. In line 48, reference # 9 reported that demodicosis had no significant relevancy with dry eye.
  4. In line 77, it is unclear that all ophthalmic examinations were performed before cataract surgery.
  5. In lines 80-81, the Schirmer I test and the BUT test must be performed before and after cataract surgery.
  6. In lines 105-107, the results should be showed in a Figure or a Table. Used product names should be described.
  7. In lines 136-137, the results should be showed in a Figure or a Table. The meaning of “between the examinations” is unclear. Comparison between pre-operation and post-operation must be performed.
  8. In lines 138-140, the results should be showed in a Figure or a Table. This result suggested demodicosis might impair the tear film.
  9. In lines 141-142, the results should be showed in a Figure or a Table. It is unclear whether break up time was extended or shortened. If it was extended, demodicosis might improve the tear film. If it was shortened, demodicosis might impair the tear film.
  10. In lines 147-148, the sentence is the repetition of the results. Please discuss.
  11. In lines 159-161, the sentence is the repetition of the results. Please discuss. Why the difference from the results of reference 22 was occurred?
  12. In lines 163-164, the sentence is the repetition of the results. Please discuss. Why the difference from the results of previous reports was occurred?
  13. In lines 165-175, it is unclear the relationship between these previous reports and the present study. Please discuss the relevance to the present study.
  14. In lines 185-186, why Demodex mites reduced the results of the Schirmer test in the present study in contrast to the previous study? Please discuss.
  15. In lines 186-188, the meaning of “between the Schirmer and BUT test in the 3-month postoperative period” is unclear. Did you compare the values of the Schirmer I test and the BUT test? If you want to examine the influence of the cataract surgery on tear film and homeostasis of ocular surface, you must compare the values before surgery and the values after surgery in these tests. The authors concluded that cataract surgery did not impair tear film and homeostasis of ocular surface in contrast to the previous studies. Do you think that no impairment of tear film after cataract surgery is general phenomenon? Is there any factor to prevent the impairment in the operations performed in the present study? Please describe the detailed methods of cataract operation and used medications in Material and Methods. Additionally, the changes in the number of Demodex mites after surgery should be described in Results.
  16. In lines 192-194, the sentence is the repetition of the results. Please discuss. Additionally, please describe the meaning of “control group” in Material and Methods.
  17. In lines 196-199, please discuss the reason of low prevalence.
  18. In lines 215-216, the results (lines 138-142) suggested that infection of Demodex mites impaired the tear film.

Author Response

In this study, the authors analyzed the prevalence of ocular demodicosis and ocular surface conditions in patients selected for cataract surgery. The results showed the presence of Demodex mites influences the conjunctiva and lid margins leading to inflammation, but does not disturb the tear film. Because a healthy ocular surface is crucial for the best outcome in cataract surgery, the influence of infection of Demodex mites on the ocular surface condition after cataract surgery is quite important.

General comments:

It is unclear what the novelty of this study is. The relationship between demodicosis and eye symptoms including ocular surface condition in normal (pre-operative) eyes has been previously reported. It may be important whether the demodicosis affects the eye condition after cataract surgery or not.

In the abstract (line 31-33) it is now added:

Schirmer I and BUT test results were lower in patients with Demodex infestation before and after cataract surgery. The higher number of mites was correlated with lower Schirmer I test results postoperatively.

And line 34-35: The higher number of Demodex mites disturbs the tear film over time after cataract surgery.

In the discussion chapter (line 371-377) it has been added:

In our study there were significant differences in the Schirmer I and BUT tests between patients with Demodex infestation and without in the 3-month postoperative period. Thus, cataract surgery impaired significantly the tear film and homeostasis of ocular surface in patients with Demodex infestation. It is already known that there is a substantial decrease in the BUT test approximately 3 months after cataract surgery [44], thus Demodex infestation can exaggerate the symptoms of dry eye syndrome after cataract surgery. Patients should be examined before cataract surgery in regard to Demodex infestation and informed about dry symptoms postoperatively.

It is unclear when ophthalmic examinations were performed. Although the authors described that there were no significant differences in the BUT and Schirmer I tests between examinations pre- and postoperatively in Abstract, no comparison between pre-operation and post-operation was described in Material and Methods and Results. To examine the influence of ocular demodicosis on the eye condition after cataract surgery, all results of ophthalmic examinations must be compared between pre-operation and post-operation.

In the Material and method section (line 101-108), the following sentence is written: The patients were examined with the use of the slit lamp before surgery and postoperatively after one and three months.  The following parameters of the anterior segment were checked preoperatively: hyperemia of the conjunctiva, blepharitis (teleangiectasia of the lid margin), loss of lashes, discharge on the lid margins, and defects of epithelium of the cornea.

Additionally, the Schirmer I test (without anesthesia with the eyes closed for 2 or 5 min.) and the tear film break up time (BUT) test were assessed at each visit (preoperatively, after one and three months postoperatively).

Additionally, comparison between patients with demodicosis and ones without demodicosis should be done before and after surgery, respectively.

In the Results section (line 216-252), the following tables have been added:

Table 7.

Values of Schirmer test I results (mm) during the follow-up in a group of patients with Demodex infestation and without (SD - standard deviation).

Visit

Demodex

No Demodex

Statistical analysis

Mean

Median

SD

Mean

Median

SD

Z

p

Preoperatively

12.10

10.50

8.43

16.76

16.25

9.37

2.07

0.04*

After one month

11.59

9.50

9.40

16.09

13.50

9.43

2.25

0.02*

After 3 months

12.03

10.50

7.98

16.12

14.25

7.90

2.32

0.02*

Table 8.

Values of the BUT test results (sec) during the follow-up in a group of patients with Demodex infestation and without (SD - standard deviation).

Visit

Demodex

No Demodex

Statistical analysis

Mean

Median

SD

Mean

Median

SD

Mean

Median

Preoperatively

7.46

5.50

4.29

9.96

9.50

5.56

2.06

0.04*

After one month

6.70

5.50

4.17

8.32

8.00

3.62

2.37

0.02*

After 3 months

6.21

5.00

4.46

7.28

7.00

3.24

1.92

0.05

As the authors indicated, many studies reported deterioration of the tear film after cataract surgery. It is unclear why cataract surgery did not impair tear film and homeostasis of ocular surface in this study. If cataract surgery does not impair ocular surface regardless of the infection of Demodex mites, it is impossible to access the influence of demodicosis on the ocular surface condition after cataract surgery. Although the authors argued that the presence of Demodex mites does not disturb the tear film, the results in this study demonstrated the significant relationship between the number of mites and results of tear film test (Schirmer I test and BUT test). It is difficult to understand what the authors mean.

In the abstract (line31-35) the following sentence has been written: Schirmer I and BUT test results were lower in patients with Demodex infestation before and after cataract surgery. The higher number of mites was correlated with lower Schirmer I test results postoperatively. The presence of Demodex mites influences the conjunctiva and lid margins leading to inflammation. The higher number of Demodex mites disturbs the tear film over time after cataract surgery.

In the Discussion section (line 371-377), the following sentences have been added:

In our study there were significant differences in the Schirmer I and BUT tests between patients with Demodex infestation and without in the 3-month postoperative period. Thus, cataract surgery impaired significantly the tear film and homeostasis of ocular surface in patients with Demodex infestation. It is already known that there is a substantial decrease in the BUT test approximately 3 months after cataract surgery [44], thus Demodex infestation can even exaggerate the symptoms of dry eye syndrome after cataract surgery. Patients should be examined before cataract surgery in regard to Demodex infestation and informed about dry symptoms that can evolve postoperatively.

The use of artificial tears and tissues should have a large impact on the results of this study. The patients in this study should be divided by the use of artificial tears and tissues.

In the results section (line 145-148), it is written: In the group of patients with the confirmed presence of Demodex, patients who reported using artificial tears (Systane, Alcon) and tissues (Blephaclean, Thea) for cleaning the lid margins were found to have greater numbers of mites: more than two mites were found in 37% of these patients and only in 6% who were not using tears or tissues (p < 0.00001).

The following table has been added (149-151):

Table 1. Relationship between the number of parasites found in one sample and the use of

artificial tears or tissues.

Artificial tears or tissues

Number of Demodex mites

Total

n (%)

0

n (%)

1-2

n (%)

> 2

n (%)

Yes

0

17

10

27

0,00%

62,96%

37,04%

100,00%

No

38

5

3

46

82,61%

10,87%

6,52%

100,00%

Total

38

22

13

73

52,05%

30,14%

17,81%

100,00%

Chi2=46,52; p<0,00001*

*the results of statistical analysis.

In the discussion chapter (line 539-541)  it has been added:

Lower prevalence of Demodex infestation in our study may be explained by the fact that patients selected for the cataract surgery take more attention to the lid hygiene, than normal population of patients at this age.

Descriptions in Discussions tend to repetition of the results. The authors should discuss the meaning of the results.

In the discussion chapter (line 371-377)  it has been added:

In our study there were significant differences in the Schirmer I and BUT tests between patients with Demodex infestation and without in the 3-month postoperative period. Thus, cataract surgery impaired significantly the tear film and homeostasis of ocular surface in patients with Demodex infestation. It is already known that there is a substantial decrease in the BUT test approximately 3 months after cataract surgery [44], thus Demodex infestation can even exaggerate the symptoms of dry eye syndrome after cataract surgery. Patients should be examined before cataract surgery in regard to Demodex infestation and informed about dry symptoms that can evolve postoperatively.

Line 381-525: In our research, the number of Demodex mites was similar in the group of patients with and without the lack of lashes.  The reason may be that the number of lashes was not really counted, assessment was done only subjectively by the examiner.

Line 537-541: Lower prevalence of Demodex infestation in our study may be explained by the fact that patients selected for the cataract surgery take more attention to the lid hygiene, than normal population of patients at this age.

Specific comments:

In lines 31-32, the comparison between pre-operation and post-operation is not described in Material and Methods and Results.

In the Material and Methods section (line 82-84), the following sentence is written: Each patient had one eye examined before the surgery and after one and three months postoperatively at the Department of Ophthalmology, John Paul II Public Hospital in Zamość, Poland.

In the Results section (line 216-252), the following tables have been added:

Table 7.

Values of Schirmer test I results (mm) during the follow-up in a group of patients with Demodex infestation and without (SD - standard deviation).

Visit

Demodex

No Demodex

Statistical analysis

Mean

Median

SD

Mean

Median

SD

Z

p

Preoperatively

12.10

10.50

8.43

16.76

16.25

9.37

2.07

0.04*

After one month

11.59

9.50

9.40

16.09

13.50

9.43

2.25

0.02*

After 3 months

12.03

10.50

7.98

16.12

14.25

7.90

2.32

0.02*

Table 8.

Values of the BUT test results (sec) during the follow-up in a group of patients with Demodex infestation and without (SD - standard deviation).

Visit

Demodex

No Demodex

Statistical analysis

Mean

Median

SD

Mean

Median

SD

Mean

Median

Preoperatively

7.46

5.50

4.29

9.96

9.50

5.56

2.06

0.04*

After one month

6.70

5.50

4.17

8.32

8.00

3.62

2.37

0.02*

After 3 months

6.21

5.00

4.46

7.28

7.00

3.24

1.92

0.05

In lines 32-33, the results (lines 138-142) suggested that infection of Demodex mites impaired the tear film.

In the abstract (line31-35) the following sentence has been written: Schirmer I and BUT test results were lower in patients with Demodex infestation before and after cataract surgery. The higher number of mites was correlated with lower Schirmer I test results postoperatively. The presence of Demodex mites influences the conjunctiva and lid margins leading to inflammation. The higher number of Demodex mites disturbs the tear film over time after cataract surgery.

In the Discussion section (line 371-377), the following sentences have been added:

In our study there were significant differences in the Schirmer I and BUT tests between patients with Demodex infestation and without in the 3-month postoperative period. Thus, cataract surgery impaired significantly the tear film and homeostasis of ocular surface in patients with Demodex infestation. It is already known that there is a substantial decrease in the BUT test approximately 3 months after cataract surgery [44], thus Demodex infestation can even exaggerate the symptoms of dry eye syndrome after cataract surgery. Patients should be examined before cataract surgery in regard to Demodex infestation and informed about dry symptoms that can evolve postoperatively.

In line 48, reference # 9 reported that demodicosis had no significant relevancy with dry eye.

As in the reference 9 blepharitis is mentioned as the reason for blepharitis, not dry eye, this reference was moved to  “blepharitis” part of the sentence (line 59-60). Ophthalmic demodicosis manifests as persistent blepharitis [9], chalazion, and dry eye syndrome [10] or may be associated with eyelid basal cell carcinoma [11].

In line 77, it is unclear that all ophthalmic examinations were performed before cataract surgery.

The following sentences are written in the Material and Methods section (line 101-108):

The patients were examined with the use of the slit lamp before surgery and postoperatively after one and three months.  The following parameters of the anterior segment were checked preoperatively: hyperemia of the conjunctiva, blepharitis (teleangiectasia of the lid margin), loss of lashes, discharge on the lid margins, and defects of epithelium of the cornea.

Additionally, the Schirmer I test (without anesthesia with the eyes closed for 2 or 5 min.) and the tear film break up time (BUT) test were assessed at each visit (preoperatively, after one and three months postoperatively) both in the group with Demodex infestation and without Demodex infestation (control group).

In lines 80-81, the Schirmer I test and the BUT test must be performed before and after cataract surgery.

In the Material and Methods section, the following sentence has been extended (line 105-108): Additionally, the Schirmer I test (without anesthesia with the eyes closed for 2 or 5 min.) and the tear film break up time (BUT) test were assessed at each visit (preoperatively, after one and three months postoperatively) both in the group with Demodex infestation and without Demodex infestation (control group).

In lines 105-107, the results should be showed in a Figure or a Table. Used product names should be described.

The names of products and table have been added in the results section (lines 145-151):

In the group of patients with the confirmed presence of Demodex, patients who reported using artificial tears (Systane, Alcon) and tissues (Blephaclean, Thea) for cleaning the lid margins were found to have greater numbers of mites: more than two mites were found in 37% of these patients and only in 6% who were not using tears or tissues (p < 0.00001) (Table 1).

Table 1. Relationship between the number of parasites found in one sample and the use of

artificial tears or tissues.

Artificial tears or tissues

Number of Demodex mites

Total

n (%)

0

n (%)

1-2

n (%)

> 2

n (%)

Yes

0

17

10

27

0,00%

62,96%

37,04%

100,00%

No

38

5

3

46

82,61%

10,87%

6,52%

100,00%

Total

38

22

13

73

52,05%

30,14%

17,81%

100,00%

Chi2=46,52; p<0,00001*

*the results of statistical analysis.

In lines 136-137, the results should be showed in a Figure or a Table. The meaning of “between the examinations” is unclear. Comparison between pre-operation and post-operation must be performed.

In the Results section (line 216-252), the following tables have been added:

Table 7.

Values of Schirmer test I results (mm) during the follow-up in a group of patients with Demodex infestation and without (SD - standard deviation).

Visit

Demodex

No Demodex

Statistical analysis

Mean

Median

SD

Mean

Median

SD

Z

p

Preoperatively

12.10

10.50

8.43

16.76

16.25

9.37

2.07

0.04*

After one month

11.59

9.50

9.40

16.09

13.50

9.43

2.25

0.02*

After 3 months

12.03

10.50

7.98

16.12

14.25

7.90

2.32

0.02*

Table 8.

Values of the BUT test results (sec) during the follow-up in a group of patients with Demodex infestation and without (SD - standard deviation).

Visit

Demodex

No Demodex

Statistical analysis

Mean

Median

SD

Mean

Median

SD

Mean

Median

Preoperatively

7.46

5.50

4.29

9.96

9.50

5.56

2.06

0.04*

After one month

6.70

5.50

4.17

8.32

8.00

3.62

2.37

0.02*

After 3 months

6.21

5.00

4.46

7.28

7.00

3.24

1.92

0.05

In lines 138-140, the results should be showed in a Figure or a Table. This result suggested demodicosis might impair the tear film.

The following table has been added (line 263-276):

Table 9. Correlation between the number of mites and the results of the Schirmer I test at each visit (preoperatively, postoperatively after one month and after 3 months)

Visit

R

p

Preoperatively

-0,22

0,06

After one month

-0,24

0,04*

After 3 months

-0,25

0,03*

In lines 141-142, the results should be showed in a Figure or a Table. It is unclear whether break up time was extended or shortened. If it was extended, demodicosis might improve the tear film. If it was shortened, demodicosis might impair the tear film.

The following tables have been added (line 233-252):

Table 8.

Values of the BUT test results (sec) during the follow-up in a group of patients with Demodex infestation and without (SD - standard deviation).

Visit

Demodex

No Demodex

Statistical analysis

Mean

Median

SD

Mean

Median

SD

Mean

Median

Preoperatively

7.46

5.50

4.29

9.96

9.50

5.56

2.06

0.04*

After one month

6.70

5.50

4.17

8.32

8.00

3.62

2.37

0.02*

After 3 months

6.21

5.00

4.46

7.28

7.00

3.24

1.92

0.05

Line 282-293:

Table 10. Correlation between the number of mites and the results of the BUT test at each visit (preoperatively, postoperatively after one month and after 3 months)

Visit

R

p

Preoperatively

-0.30

0.01

After one month

-0.31

0.01

After 3 months

-0.26

0.02

* Spearman's R coefficient

In the abstract (line 31-35) it is now written as follows:

Schirmer I and BUT test results were lower in patients with Demodex infestation before and after cataract surgery. The higher number of mites was correlated with lower Schirmer I test results postoperatively. The presence of Demodex mites influences the conjunctiva and lid margins leading to inflammation. The higher number of Demodex mites disturbs the tear film over time after cataract surgery.

As conclusion (line 549-551) it is now written:

Demodex folliculorum infestation is a common condition in patients selected for cataract surgery. The higher number of Demodex mites influences the conjunctiva and lid margin and leads to inflammation and disturbance of the tear film.

In lines 147-148, the sentence is the repetition of the results. Please discuss.

The following sentence has been removed from the discussion chapter (line 298):

73 patients.

Additionally,in the discussion chapter it has been added (line 298-302):

The possible reason may be that D. folliculorum can be more easily isolated than D. brevis [20], as D. folliculorum exists in the lash follicle, whereas D. brevis the lash's sebaceous gland and the meibomian gland [2]. Thus, D. folliculorum is more commonly seen in posterior blepharitis, or keratoconjunctivitis and D. brevis is more common in the sebaceous gland- or meibomian gland-related diseases, such as chalazion [15].

The following reference has been added:

  1. Zeytun, E.; Karakurt, Y. Prevalence and load of Demodex folliculorum and Demodex brevis (Acari: Demodicidae) in patients with chronic blepharitis in the province of Erzincan, Turkey. Journal of medical entomology. 2019, 56(1), 2–9.In lines 159-161, the sentence is the repetition of the results.

The following sentence has been removed from the discussion:

In our study, the result was very similar, as 72% of the patients with blepharitis had positive result for the presence of Demodex but there was no significant relationship of the Demodex infestation with age and gender.

Why the difference from the results of reference 22 was occurred?

Most of the studies are in agreement with our results – Demodex infestation is more prevalent in patients with blepharitis.

In the discussion chapter (line 333-339)  it is written:

Most authors demonstrate a higher prevalence of Demodex mites in patients with blepharitis compared to healthy controls [9,21,22], which in accordance with our study, whereas some authors show a similar prevalence of Demodex mites in blepharitis and control groups. Kemal found Demodex in 28.8% (49/170) of patients with blepharitis and in 26.7% (88/330) of controls [23]. The difference between the two groups was not statistically significant and there was no relationship between the presence of D. folliculorum and host factors (age, sex).

In lines 163-164, the sentence is the repetition of the results. Please discuss. Why the difference from the results of previous reports was occurred?

In the discussion chapter (line 345in our study-348) it is written:

It may be due to the fact that a majority of our patients were at the age of 70-79 years, and there were more females than males in the study group, this profile is typical for patients who undergone cataract surgery.

In lines 165-175, it is unclear the relationship between these previous reports and the present study. Please discuss the relevance to the present study.

The following sentence has been moved to the section of discussion dealing with blepharitis in Demodex infestation (line 304-333): Intensive D. folliculorum invasions cause keratinization, hyperplasia, distension, and melanocyte aggregation. Large populations of D. brevis may destroy glandular cells, produce granuloma, and plug the ducts of the Meibomian or sebaceous glands [28].

In lines 185-186, why Demodex mites reduced the results of the Schirmer test in the present study in contrast to the previous study? Please discuss. In lines 186-188, the meaning of “between the Schirmer and BUT test in the 3-month postoperative period” is unclear. Did you compare the values of the Schirmer I test and the BUT test?

In the Discussion section (line 371-377:), the following sentences have been added:

In our study there were significant differences in the Schirmer I and BUT tests between patients with Demodex infestation and without in the 3-month postoperative period. Thus, cataract surgery impaired significantly the tear film and homeostasis of ocular surface in patients with Demodex infestation. It is already known that there is a substantial decrease in the BUT test approximately 3 months after cataract surgery [44], thus Demodex infestation can even exaggerate the symptoms of dry eye syndrome after cataract surgery. Patients should be examined before cataract surgery in regard to Demodex infestation and informed about dry symptoms that can evolve postoperatively.

If you want to examine the influence of the cataract surgery on tear film and homeostasis of ocular surface, you must compare the values before surgery and the values after surgery in these tests.

In the results section (209-252) it has been added:

There were significant differences both in Schirmer I (Table 7) and BUT test (Table 8) between patients with Demodex infestation and without. Schirmer test results were lower in patients with Demodex infestation. BUT test was significantly shorter in Demodex positive patients.

The statistical analysis did not reveal any significant differences in the Schirmer I and BUT test results between the examinations pre- and postoperatively both in the group with Demodex infestation and without (Table 7 and 8).

Table 7.

Values of Schirmer test I results (mm) during the follow-up in a group of patients with Demodex infestation and without (SD - standard deviation).

Visit

Demodex

No Demodex

Statistical analysis

Mean

Median

SD

Mean

Median

SD

Z

p

Preoperatively

12.10

10.50

8.43

16.76

16.25

9.37

2.07

0.04*

After one month

11.59

9.50

9.40

16.09

13.50

9.43

2.25

0.02*

After 3 months

12.03

10.50

7.98

16.12

14.25

7.90

2.32

0.02*

Table 8.

Values of the BUT test results (sec) during the follow-up in a group of patients with Demodex infestation and without (SD - standard deviation).

Visit

Demodex

No Demodex

Statistical analysis

Mean

Median

SD

Mean

Median

SD

Mean

Median

Preoperatively

7.46

5.50

4.29

9.96

9.50

5.56

2.06

0.04*

After one month

6.70

5.50

4.17

8.32

8.00

3.62

2.37

0.02*

After 3 months

6.21

5.00

4.46

7.28

7.00

3.24

1.92

0.05

The authors concluded that cataract surgery did not impair tear film and homeostasis of ocular surface in contrast to the previous studies. Do you think that no impairment of tear film after cataract surgery is general phenomenon? Is there any factor to prevent the impairment in the operations performed in the present study?

The results and conclusions in regards to the tear film have been changed (see earlier corrections).

Please describe the detailed methods of cataract operation and used medications in Material and Methods.

In the Material and Methods section (line 85-89), the following sentences have been added:

The cataract surgeries were performed by the same experienced surgeon in a standard manner after topical anesthesia with proparacaine hydrochloride 0.5%. After making a 2.2-mm clear corneal incision, continuous capsulorrhexis, hydrodyssection, and phacoemulsification were performed (Infiniti Vision System Alcon, US) and the IOL was inserted into the capsular bag. All surgeries were performed without complications.

Additionally, the changes in the number of Demodex mites after surgery should be described in Results.

The number of Demodex mites was checked only once-preoperatively, the correlation was found between the number of mites and BUT and Schirmer tests preoperatively and after 1 and 3 months.

The following text and tables have been added (line 259-293):

A correlation was found between the preoperative number of mites and the results of the Schirmer I test after one month (R = -0.24, p = 0.04) and after 3 months (R = -0.25, p = 0.03) (table 9). The higher number of mites was correlated with lower Schirmer I test results.

Table 9. Correlation between the number of mites and the results of the Schirmer I test at each visit (preoperatively, postoperatively after one month and after 3 months)

Visit

R*

p

Preoperatively

-0.22

0.06

After one month

-0.24

0.04

After 3 months

-0.25

0.03

* Spearman's R coefficient

There was also a significant relationship between the results of BUT and the number of mites after one month (R = -0.31, p = 0.01) and 3 months (R = -0.26, p = 0.02) postoperatively (table 10).

Table 10. Correlation between the number of mites and the results of the BUT test at each visit (preoperatively, postoperatively after one month and after 3 months)

Visit

R

p

Preoperatively

-0.30

0.01

After one month

-0.31

0.01

After 3 months

-0.26

0.02

* Spearman's R coefficient

In lines 192-194, the sentence is the repetition of the results. Please discuss.

In the discussion (line 381-525) it is now written: In our research, the number of Demodex mites was similar in the group of patients with and without the lack of lashes.

The reason may be that the number of lashes was not really counted, assessment was done only subjectively by the examiner.

Additionally, please describe the meaning of “control group” in Material and Methods.

In Material and methods section (line 105-108) the following sentence is written:

Additionally, the Schirmer I test (without anesthesia with the eyes closed for 2 or 5 min.) and the tear film break up time (BUT) test were assessed at each visit (preoperatively, after one and three months postoperatively) both in the group with Demodex infestation and without Demodex infestation (control group).

In the Discussion chapter, the following sentence (line 193-195): In our research, the lack of eyelashes in patients with diagnosed D. folliculorum invasion was similar to that of the control group.

has been changed into (line 381-525):

In our research, the number of Demodex mites was similar in the group of patients with and without the lack of lashes.

The reason may be that the number of lashes was not really counted, assessment of lack of lasheswas done only subjectively by the examiner.

In lines 196-199, please discuss the reason of low prevalence.

In the discussion chapter (line 531-535) it has been added:

However, there are some studies with similar prevalence, for example 40.2% of patients suffering from ocular discomfort [49]. Relatively lower prevalence in our study may be explained by the fact that patients selected for the cataract surgery take more attention to the lid hygiene, than normal population of patients at this age.

The following reference has been added:

  1. Rabensteiner, D.F.; Aminfar, H.; Boldin, I. et al. Demodex Mite Infestation and its Associations with Tear Film and Ocular Surface Parameters in Patients with Ocular Discomfort. Am J Ophthalmol. 2019, 204, 7-12.

In lines 215-216, the results (lines 138-142) suggested that infection of Demodex mites impaired the tear film.

The results and discussion in regard to tear film has been changed and already discussed above.

The following reference has been added:

  1. Li, X.-M.; Hu, L.; Hu, J.; Wang, W. Investigation of dry eye disease and analysis of the pathogenic factors in patients after cataract surgery. Cornea. 2007, 26:S16–S20.

Round 2

Reviewer 2 Report

There are no comments.